# Branched actin networks are organized for asymmetric force production during clathrin-mediated endocytosis in mammalian cells

Meiyan Jin [1,6], Cyna Shirazinejad [1,2,6], Bowen Wang[3], Amy Yan [1], Johannes Schöneberg[1,5], Srigokul Upadhyayula [1,4], Ke Xu [3] & David G. Drubin [1✉]

Actin assembly facilitates vesicle formation in several trafficking pathways, including clathrin-mediated endocytosis (CME). Interestingly, actin does not assemble at all CME sites in mammalian cells. How actin networks are organized with respect to mammalian CME sites and how assembly forces are harnessed, are not fully understood. Here, branched actin network geometry at CME sites was analyzed using three different advanced imaging approaches. When endocytic dynamics of unperturbed CME sites are compared, sites with actin assembly show a distinct signature, a delay between completion of coat expansion and vesicle scission, indicating that actin assembly occurs preferentially at stalled CME sites. In addition, N-WASP and the Arp2/3 complex are recruited to one side of CME sites, where they are positioned to stimulate asymmetric actin assembly and force production. We propose that actin assembles preferentially at stalled CME sites where it pulls vesicles into the cell asymmetrically, much as a bottle opener pulls off a bottle cap.

[1] Department of Molecular and Cell Biology, University of California, Berkeley, CA, USA. [2] Biophysics Graduate Group, University of California Berkeley, Berkeley, CA, USA. [3] Department of Chemistry, University of California, Berkeley, CA, USA. [4] Chan Zuckerberg Biohub, San Francisco, CA, USA. [5] Present address: Department of Pharmacology, and Department of Chemistry and Biochemistry, University of California, San Diego, CA, USA. [6] These authors contributed equally: Meiyan Jin, Cyna Shirazinejad. ✉email: drubin@berkeley.edu

Formation of clathrin-coated vesicles requires forces to first bend the membrane into a sphere or tube, and to then break the thin neck that connects the vesicle to the plasma membrane. These forces are generated through the combined actions of proteins that directly bend the membrane and actin filament assembly[1–5] (Supplementary Fig. 1a). Several studies have demonstrated that dependence of CME on actin assembly increases under elevated membrane tension[6–10]. Interestingly, actin does not assemble at all CME sites in mammalian cells, suggesting highly localized differences in requirement for actin assembly, the nature of which is obscure[6,11,12]. A detailed understanding of how actin forces are harnessed to aid vesicle formation and scission depends on understanding which CME sites assemble actin, where filament assembly occurs around the endocytic membrane, and when. In yeast cells, where turgor pressure is particularly high, super-resolution data suggest that actin assembles symmetrically around CME sites and indicate that actin regulators including Las17, which is yeast WASP, are present in a ring surrounding the base of the clathrin coat symmetrically[13]. On the other hand, studies on fixed mammalian cells raised the possibility that actin assembly may at least in some cases be initiated asymmetrically at clathrin coats[14,15]. However, methods used for these studies prevented analysis of large numbers of sites, and suffered from possible loss of actin filaments during unroofing and extraction of the cells. Which CME sites assemble actin, and how actin networks are organized with respect to CME sites, has not been determined systematically in a large-scale, unbiased manner, particularly in live mammalian cells. This information is essential to understanding how and why actin assembly forces are harnessed for CME.

Here, by combining fixed and live-cell imaging of triple-genome-edited, human induced pluripotent stem cells (iPSCs), and newly developed machine-learning-based computational analysis tools, we report that N-WASP and the Arp2/3 complex localize at one side of the coat and neck of invaginating endocytic sites until vesicle scission. Most importantly, by comparing recruitment dynamics of proteins from three distinct endocytic modules for over one thousand unperturbed endocytic events, we found that branched actin assembly occurs predominantly at sites that have stalled between coat expansion and vesicle scission. We propose that these branched actin networks rescue stalled CME.

## Results

### Super-resolution imaging reveals asymmetric actin distribution around endocytic sites

To investigate the physiological roles and spatiotemporal regulation of actin assembly at CME sites in mammalian cells, we applied genome-editing techniques to generate a human iPSC line (hereafter referred to as ADA cells) that co-expresses a TagRFP-T fusion of the mu subunit of the AP2 adaptor complex (AP2M1), a TagGFP2 fusion of dynamin2 (DNM2), and a HaloTag fusion of the ARPC3 subunit of the Arp2/3 complex as representatives of the CME coat, scission and actin modules, respectively[2,16,17] (Supplementary Fig. 1a, b). Previous studies showed that endogenously tagged AP2M1, DNM2 and ARPC3 can serve as reliable markers of these CME functional modules that avoid disruption of physiological spatiotemporal organization of the process as might be caused by overexpression of fluorescently labeled proteins[12,18–21]. We observed dynamic CME events on the basal plasma membrane of the genome-edited cells using Total Internal Reflection Fluorescence (TIRF) microscopy (Supplementary Fig. 1c and Supplementary Movie 1, 2). Consistent with previous studies, AP2 is recruited at early CME stages while DNM2 is recruited in two phases[11,16,20,22]. At the early stage of CME, a relatively small amount of DNM2 is recruited to CME sites. Shortly before the

end of a CME event, the DNM2 recruitment rate increases rapidly with DNM2 levels reaching a peak concomitant with vesicle scission[16,20,23] (Supplementary Fig. 1c). This later rapid-recruitment phase represents the assembly of the dynamin helix on the highly curved neck of the budding vesicle after the U to Ω shape transition of the endocytic membrane[20,23–27].

Super-resolution imaging of fixed human skin melanoma SKMEL cells suggested that actin is arranged asymmetrically around CME sites[7], consistent with indications from other previous studies[14,15]. To systematically analyze how actin networks are organized at CME sites in iPSCs, we first performed two-color 3D Stochastic Optical Reconstruction Microscopy (STORM) imaging[28] on fixed ADA cells, localizing either AF647 phalloidin-labeled actin filaments[29] or HaloTag-fused ARPC3 at CME sites. Due to the dense phalloidin labeling of cortical actin filaments under the plasma membrane, it was often challenging to unambiguously identify the CME-specific actin structures in iPSCs. However, in regions with thinner cortical actin layers, we observed that actin was typically distributed asymmetrically around CME sites (Fig. 1a–c). Antibody labeling of ARPC3-Halotag in the ADA cells had the advantage of a less complex staining pattern. Besides being highly concentrated in lamellipodia, ARPC3 was associated with CME sites asymmetrically, like actin (Fig. 1d–f). These data suggest that the Arp2/3-mediated actin network is arranged asymmetrically around CME sites.

### Actin networks assembled at CME sites remain asymmetric through scission

We next used ADA cells to investigate actin assembly at CME sites in live cells, which has several advantages over studies in fixed cells. During the fixation and subsequent sample preparation, actin structures may not be faithfully preserved. In addition, in live cells it is easier to identify the stage of CME, so the timing, geometry and dynamics of actin assembly can be related to the endocytic stage. More importantly, only by using live cells is it possible to trace a single CME event from start to finish, and to therefore identify those CME events wherein no detectable actin is ever assembled, so key parameters can be compared between events with and without associated actin assembly.

By visualizing endogenously tagged AP2M1 to mark the coat and CME initiation, and DNM2 to mark the neck and scission, together with ARPC3 to specifically label Arp2/3-nucleated, branched actin filaments (Supplementary Fig. 1a), we were able to precisely study the spatial and temporal regulation of actin assembly during CME. Three-color labeling and analysis of the displacement between markers for the three modules allowed us to distinguish bona fide asymmetric actin assembly from events that might artifactually appear asymmetric because the invaginations were elongated and tilted (Fig. 2a). Using TIRF live-cell imaging, we observed ARPC3-labeled branched actin networks at lamellipodia and at a subpopulation of CME sites (Fig. 2b, c). Dynamic actin assembly and disassembly occurred at CME sites with different spatio-temporal characteristics, including discrete CME sites, clathrin plaques and at clathrin coat splitting sites, as previously reported[14] (Fig. 2d and Supplementary Fig. 2a, b). In the analysis described below, we focus on the discrete CME events and not the more complex ones (plaques and splitting events) (Supplementary Fig. 3a). Analysis of the discrete events with 1 s/frame temporal resolution revealed that ARPC3 is most robustly recruited during the late stages of CME shortly before scission[12] (Fig. 2c, d). Interestingly, we observed clear spatial displacement between ARPC3 (actin module) and AP2 (coat module) as well as between ARPC3 and DNM2 (neck) before vesicle scission (Fig. 2d). This observation supports the conclusion that

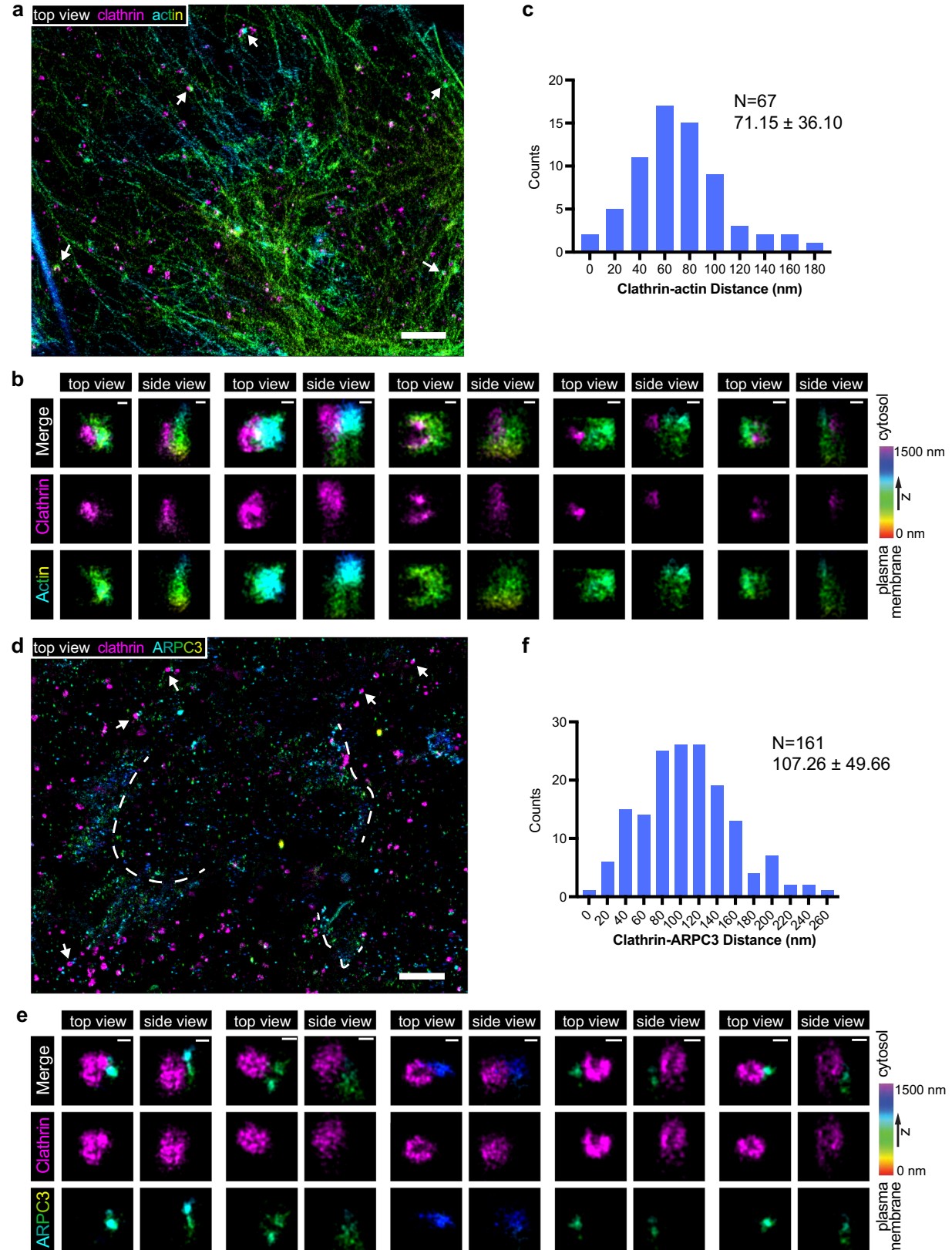

asymmetric branched actin networks provide forces at endocytic sites through the time of scission.

To investigate the spatiotemporal relationship between actin and dynamin with higher resolution, we used live-cell Airyscan2 confocal microscopy of live cells (Fig. 3 and Supplementary Movie 3). We observed clear displacement between ARPC3 and

DNM2 signals at the time of vesicle scission at most CME sites (Fig. 3b). Notably, ARPC3 remained on one side of DNM2 throughout the last second of CME until scission (Fig. 3c, d). These observations of asymmetric branched actin network assembly at CME sites are consistent with the ones we made with TIRF live-cell imaging (Fig. 2).

**Fig. 1 Two-color, 3D stochastic optical reconstruction microscopy (STORM) shows that actin structures are off-centered with respect to clathrin coats. a, b** Two-color 3D STORM image of bottom membrane of ADA cells immunolabeled with clathrin light chain antibody (clathrin, CF-680, magenta) and phalloidin (actin, AF647, rainbow). **c** A histogram of distances between centroids of clathrin and actin signals. Mean and standard deviation are reported on the graph. Data were analyzed using Prism 9. Source data are provided in the Source Data file. **d, e** Two color 3D STORM image of the bottom membrane of ADA cells immunolabeled with clathrin light chain antibody (clathrin, AF647, magenta) and HaloTag antibody (ARPC3-HaloTag, CF-680, rainbow). Dotted lines label lamellipodia. **b, e** The highlighted CME sites, which are labeled by white arrows in **a, d**, are rotated and shown in magnified top and side view projections. Color bar shows the z position of ARPC3-HaloTag. **f** A histogram of distances between centroids of clathrin and ARPC3 signals. Mean and standard deviation are reported on the graph. Data were analyzed using Prism 9. Source data are provided in the Source Data file. Scale bars: **a, d**: 2 μm, **b, e**: 100 nm.

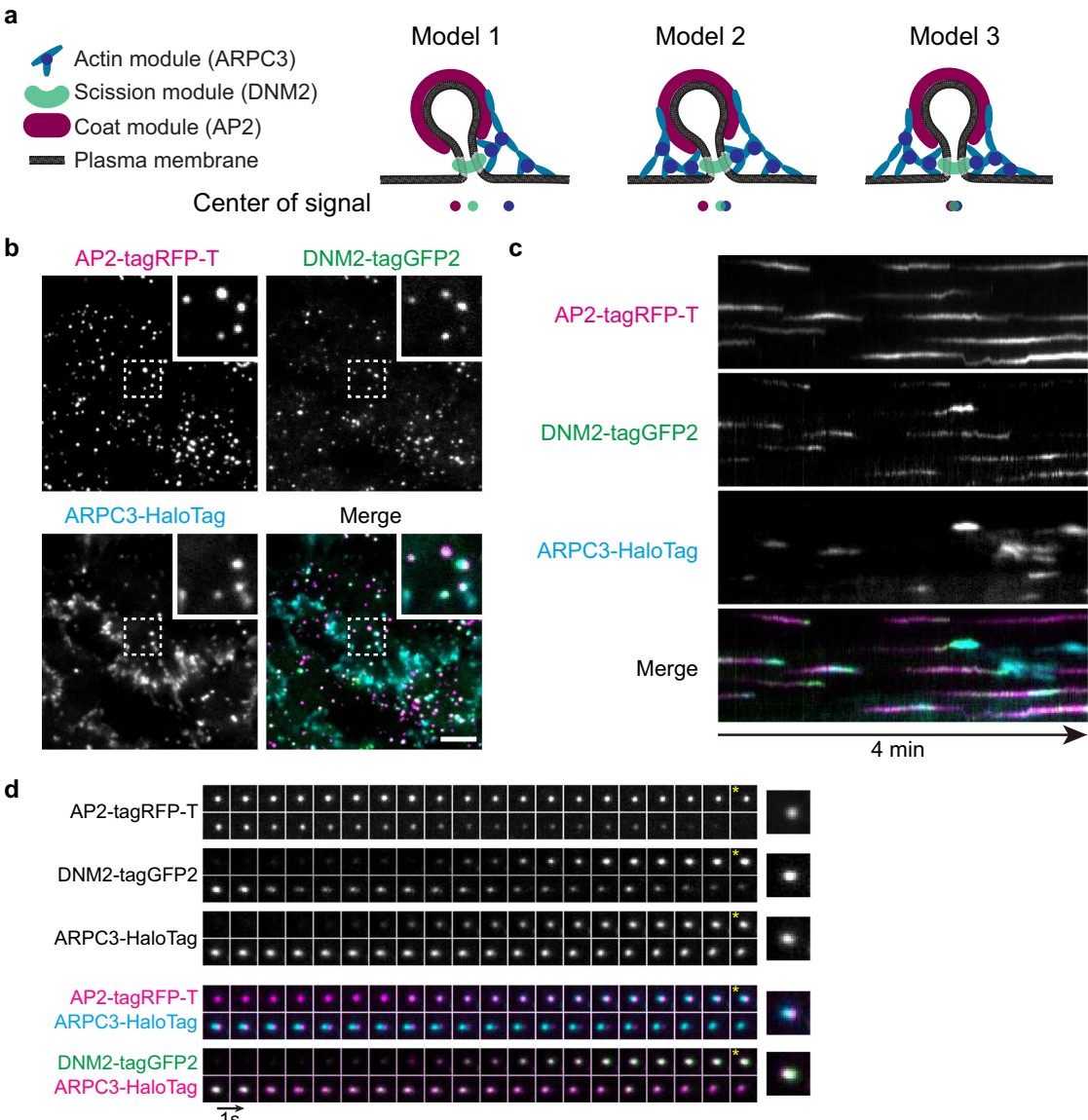

**Fig. 2 Triple-genome-edited iPSCs reveal dynamic actin organization at CME sites. a** Models of branched actin assembly at invaginating CME sites. Model 1: Asymmetric actin assembly at CME sites results in separated actin-coat and actin-neck signals. Model 2: Symmetric actin assembly at tilted CME sites results in separated actin-coat signals but overlapped actin-neck signals. Model 3: Symmetric actin assembly at perpendicularly invaginating CME sites will result overlapped actin, coat and neck signals. **b** A representative single time frame image of a TIRF movie (Supplementary Movie 2) of AP2M1-tagRFP-T (magenta), DNM2-tagGFP2 (green) and JF635 ligand[51]-conjugated ARPC3-HaloTag (cyan) in ADA cells. The highlighted region is boxed by a dashed line. Scale bar: 5 μm. **c** A representative kymograph of AP2M1-tagRFP-T (magenta), DNM2-tagGFP2 (green) and JF635 ligand-conjugated ARPC3-HaloTag (cyan) at CME sites in ADA cells. Scale bar: 5 μm. **d** Montage of a representative ARPC3 positive CME site in ADA cells. Individual channels and pair-wise merges are shown. *: Images from one frame before scission (maximum DNM2 intensity) are marked to show the displacement between the CME coat (AP2)-ARPC3 and CME neck (DNM2)-ARPC3. Size of field of view: 2 μm × 2 μm. Intervals: 1 s.

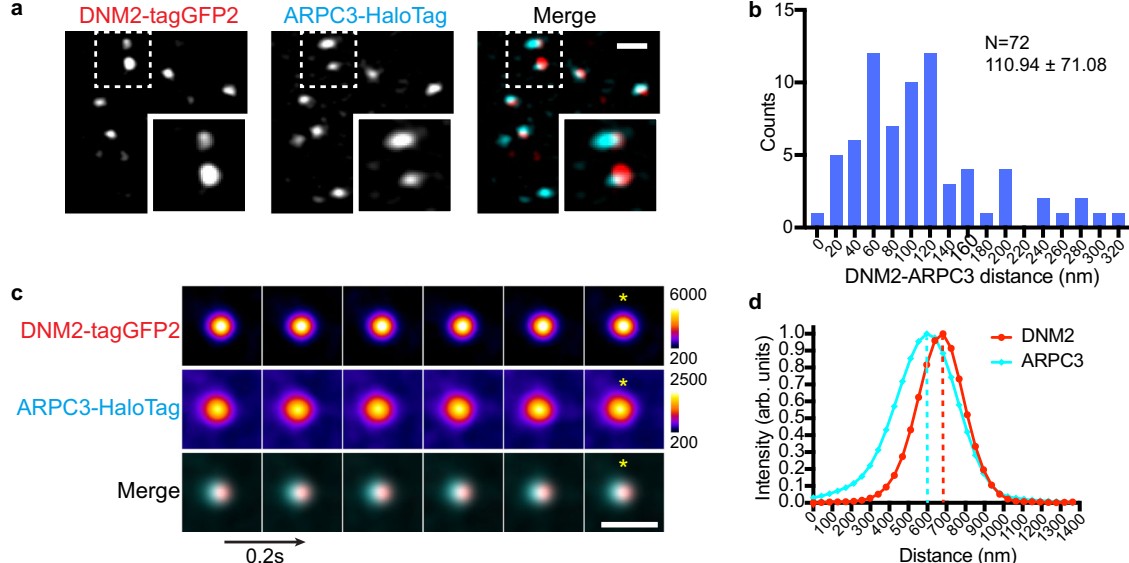

**Fig. 3 Airyscan live-cell imaging reveals asymmetric actin organization at CME sites. a** A representative single time frame image of an Airyscan movie (Supplementary Movie 3) of DNM2-tagGFP2 (red) and JF635 ligand-conjugated ARPC3-HaloTag (cyan) on the ventral plasma membrane surface of ADA cells. The highlighted region is boxed by a dashed line. Scale bar: 1 μm. **b** Histogram of distance between the center of mass of DNM2 and ARPC3 at frame corresponding to scission. Mean and standard deviation are shown on the graph. Data were analyzed using Prism 9. Source data are provided in the Source Data file. **c** Montage of average intensity projection of representative ARPC3 positive CME sites. Frame corresponding to scission was determined by maximum DNM2 intensity. 33 × 33 pixels square region centered at DNM2 maximum intensity pixel was cropped for six continuous frames ending in scission. Cropped frame series were rotated by 90 or 180 degrees as needed to align the ARPC3 signal to the left of the center of the image at the frame corresponding to scission. *: Averaged image of the frame corresponding to scission (maximum DNM2 intensity) is marked to show the displacement between the CME neck (DNM2) and branched actin (ARPC3). Scale bar: 1 μm. Intervals: 0.2 s. N = 72. **d** The line scan function in ImageJ software was used to measure the fluorescence intensity along a one pixel wide, 33 pixel long horizontal line drawn across the center of the averaged intensity image of the frame corresponding to scission* in **c**. The signals were normalized to the minimum signal along the line, and intensity was calculated as a ratio to the maximum signal along the line. Source data are provided in the Source Data file.

To analyze the intrinsic recruitment order and timing for three endocytic proteins at CME sites quantitatively and systematically, we developed an automated, high-throughput method to analyze TIRF live-cell imaging data that avoids bias because it does not involve manual selection of CME sites (see Materials and Methods). Briefly, AP2 tracks were identified using standard particle-tracking algorithms[30]. Novel filtering methods then extracted DNM2-positive events marked by one or more DNM2 burst. The AP2 and DNM2 tracks were decomposed into dynamic features describing each events' position and brightness. These features were used for clustering via unsupervised machine learning, which enabled grouping of similarly-behaved tracks (Supplementary Fig. 3a, b). DNM2-positive events were refined by a detection scheme that determined the number of DNM2 peaks using various characteristics of a single DNM2-peak: the peak height, width, and minimum peak-to-peak distance (Supplementary Fig. 3c). Events with a single DNM2 peak were analyzed as described below. The method detects low signals from endogenously tagged CME proteins, such as the low-level recruitment of DNM2 at the early CME stages, and accurately reveals the different CME stages (Supplementary Fig. 3d).

Next, the timing of actin network assembly at CME sites was determined using ARPC3 as a branched actin filament marker by analyzing over one thousand CME events. Although actin appearance early in CME has been reported[14], determining the actin assembly timing can be challenging because it is difficult to distinguish newly assembled branched actin at CME sites from the nearby cortical actin filaments or actin filaments associated with other vesicles or organelles. Also, whether actin functions during the early stage of CME has not yet been shown conclusively due to the potential side effects such as changes in

membrane tension caused by actin inhibitors. Our endogenous ARPC3 tagging and large-scale computational analysis approach sidesteps these problems. We classified CME events into three groups: one group without ARPC3 appearance, one group with ARPC3 appearance early in CME, and one group with ARPC3 appearance late in CME (Supplementary Fig. 4). We observed that in most of the ARPC3 positive events a sharply increasing ARPC3 signal appears with similar timing to the rapid-recruitment phase of DNM2 concomitant with the U to Ω membrane shape transition (Fig. 4a and Supplementary Fig. 4b). This timing is consistent with the previously proposed role for actin in membrane invagination, as studies showed that actin inhibitors block the U to Ω endocytic membrane shape transition[6,14]. In some cases we did detect ARPC3 signals at early CME stages (Supplementary Fig. 4b). To test whether random overlap between nearby actin structures and CME sites might be responsible for the apparent early actin recruitment, we generated a randomized data set by pairing ARPC3 images with AP2 and DNM2 images from an unrelated movie (Supplementary Fig. 4a). In this data set, we detected a significantly reduced fraction of ARPC3 positive events. However, early "assembly" of actin was observed in a similar fraction of CME events as in the real data set (Supplementary Fig. 4b). Based on these observations we conclude that the presence of actin early in CME is very likely due to nearby actin structures overlapping with CME sites randomly.

Our live-cell analysis allowed the timing of branched actin network assembly to be compared to the scission timing, and the spatial offset between the clathrin coat and the associated actin network to be determined. Super-resolution imaging of yeast CME sites suggested that actin and actin nucleators localize

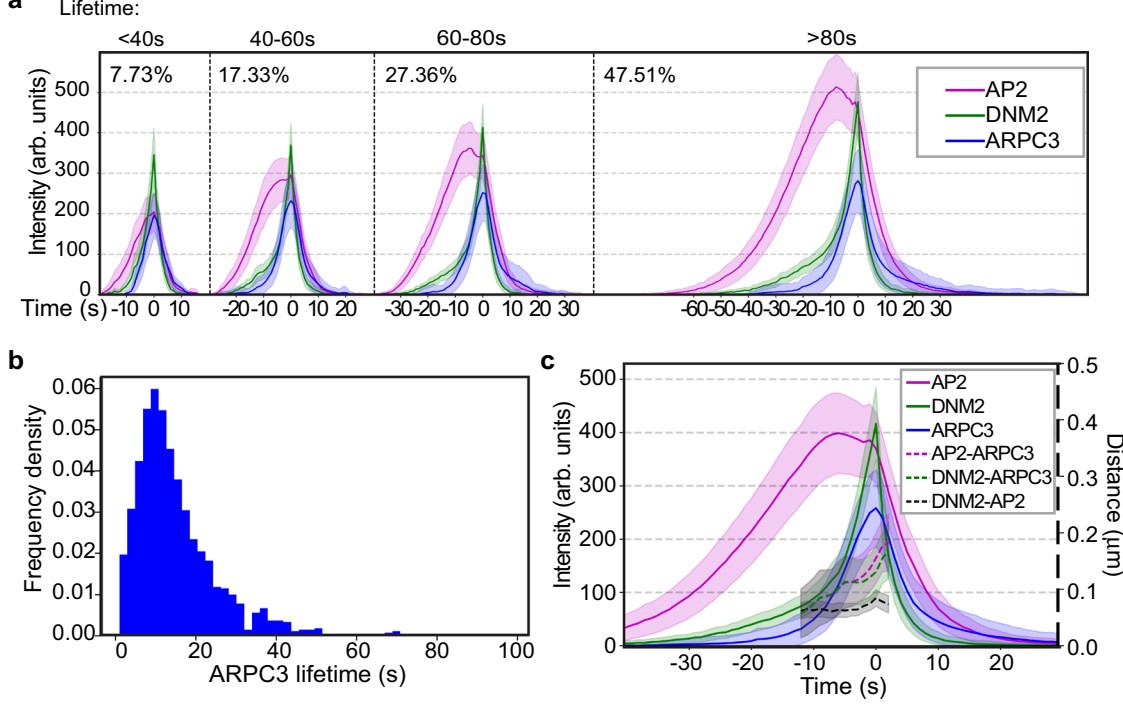

**Fig. 4 Computational analysis of ARPC3 positive CME sites reveals asymmetric actin network assembly at the late stage of CME. a** Averaged intensity vs time plots of cohorts of CME sites with late ARPC3 assembly in ADA cells. Events are grouped into cohorts by the lifetimes of AP2 and aligned to the frames showing the maximum DNM2 intensity (time = 0 s). Percentage of the number of the CME sites in each cohort is shown next to the plot. **b** Histogram of ARPC3-mediated actin network assembly duration. The assembly duration is measured from the first frame of the ARPC3 signal to the presumed scission time (the peak of DNM2 signal). Source data are provided in the Source Data file. **c** Averaged intensity (solid lines) and distance (dashed lines) vs time plots of ARPC3 positive CME sites in ADA cells. Events are aligned to the frames showing the maximum DNM2 intensity (time = 0 s). Distance between centers of two signals are shown from -10 s to 3 s when DNM2 and ARPC3 signals are relatively high. **a–c** N = 1,385. **a**, **c** Error bar: ¼ standard deviation. .

symmetrically in a ring around CME sites, and computational modeling suggested that an asymmetric actin arrangement would not provide sufficient force for the membrane invagination during yeast CME[13]. In contrast, in mammalian cells, which require less actin force production during CME, imaging of fixed cells suggested that actin structures associate adjacent to apparent flat clathrin coats[14,15]. However, these studies proposed that at the later CME stages the actin structures become larger and more symmetric to provide sufficient force for membrane deformation and scission[14,15]. Surprisingly, in our live cell studies designed to highlight sites of new actin assembly, we observed off-centered branched actin networks at CME sites throughout even the latest CME stages (Fig. 2d and Fig. 3b–d). Furthermore, most ARPC3-positive CME sites accomplish scission within 30 s of the initiation of ARPC3 recruitment (Fig. 4b). The actin networks (ARPC3) we observed were off center from the coat (AP2) and neck (DNM2) signals by approximately 150 nm at the time of vesicle scission (Fig. 4c).

Imaging fluorescent beads using the same settings indicated that the displacement is not an artifact caused by misalignment between different imaging channels (Supplementary Movie 4 and Supplementary Fig. 5a). By further analyzing fluorescent bead images, we concluded that chromatic aberration contributes only a small portion of the AP2-ARPC3 and AP2-DNM2 separation we observed by TIRF imaging (Supplementary Fig. 5b). This chromatic aberration might be the reason for slightly increased neck-actin separation we detected by TIRF microscopy (Fig. 4c) compared to Airyscan imaging (Fig. 3b). Given the temporal separation between channel acquisition and the movement of AP2 spots, it was important to assess whether the spatial

separation between channels can be attributed in part to an imaging artifact caused by puncta movement between subsequent channel acquisitions. When we measured the average movement of AP2 spots leading up to scission, we found that over 95% of the events had AP2-ARPC3 separations that exceed the frame-to-frame motility of AP2 (Supplementary Fig 5c). Also, the puncta positional uncertainties indicated by the standard deviations determined when measuring the fitted position of AP2, range up to 40 nm, which is less than the determined displacements. Therefore, we utilized the AP2-DNM2 separation, which is expected to be small, as the basis for comparison to the AP2-ARPC3 and DNM2-ARPC3 separations (Fig. 4d). These results further support our conclusion that branched actin networks assemble asymmetrically at CME sites through the time of scission (Fig. 2d). This observation is consistent with the observation that ring-shaped actin structures at clathrin coats were rarely observed in the high-resolution, live-cell imaging reported in a previous study[31]. In total, these live-cell data suggest that in mammalian cells, asymmetric actin network assembly can provide enough force to assist membrane deformation and scission during the late stages of CME.

**Asymmetric branched actin networks facilitate CME at stalled sites.** To gain additional insights into the function of this asymmetric actin network assembly, we quantitatively compared kinetics of CME events with or without ARPC3 recruitment. We observed that about 30% of CME events are completed in the absence of detectable actin assembly (Supplementary Fig. 4b), which is consistent with the hypothesis that in mammalian cells

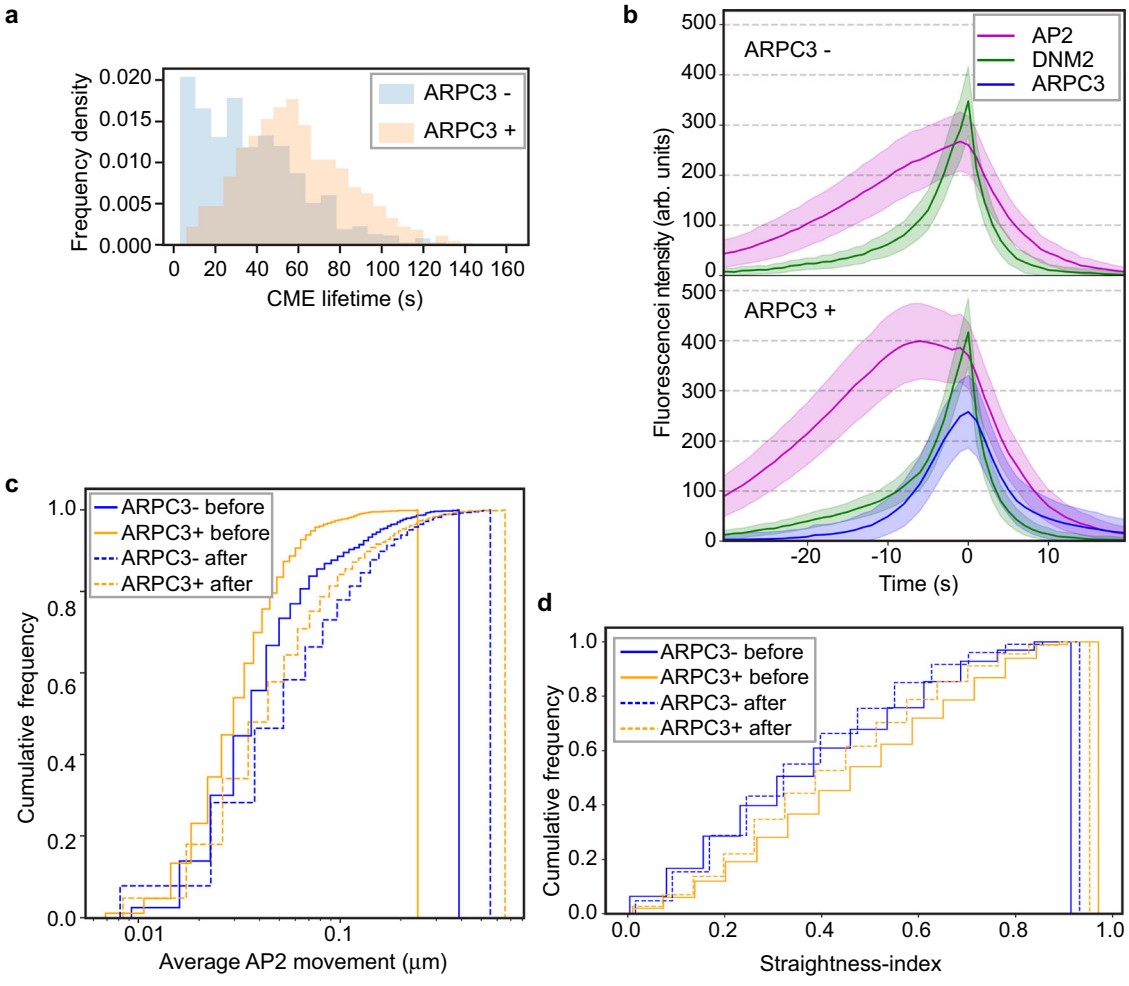

**Fig. 5 Actin positive CME sites show distinct dynamics. a** Histograms of ARPC3 negative (blue) and positive (orange) CME lifetimes. CME lifetime is measured from the first frame of the AP2 signal to the presumed scission time (the peak of DNM2 signal). ARPC3 positive CME events have longer lifetimes. p-value from two-sided Welch's t-test: 2.11e-76. **b** Averaged intensity vs time plots of ARPC3 negative (top) and positive (bottom) CME sites in ADA cells. Events were aligned to the frames showing the maximum DNM2 intensity. Error bar: ¼ standard deviation. **c** Lateral motility of ARPC3 negative (blue) and positive (yellow) CME sites before (solid line) and after (dashed line) vesicle scission. ARPC3 positive CME sites move slower than ARPC3 negative ones. p-value from two-sided Kolmogorov-Smirnov test: ARPC3 + before vs ARPC3- before: 1.27e-14, ARPC3 + after vs ARPC3- after: 5.18e-08. **d** Straightness-index of ARPC3 negative (blue) and positive (yellow) CME sites before (solid line) and after (dashed line) scission. The straightness-index is defined by the ratio between the sum of frame-to-frame distances to the end-to-end distance of a single event's trajectory, where a perfectly straight-lined trajectory would have an index of 1. APRC3 positive CME sites move with a more straight trajectory. p-value from two-sided Kolmogorov–Smirnov test: ARPC3 + before vs ARPC3- before: 1.07e-11, ARPC3 + after vs ARPC3- after: 2.06e-6. **a–d** ARPC3 -: N = 840, ARPC3 + : N = 1,385. **a**, **c**, **d** Source data are provided in the Source Data file.

actin assembly is required for CME only under certain conditions, such as relatively high membrane tension or specific cargo internalization, which can vary regionally within cells[6,7,9,10,32,33]. Consistent with the possibility that unfavorable conditions such as increased membrane tension, might stall membrane deformation during CME[4,6,8–10,34–36], CME lifetimes were markedly longer for ARPC3 positive events compared to the ARPC3 negative events (Fig. 5a). In addition, when the AP2M1 intensity vs time profiles were compared between ARPC3 positive and negative CME sites, a plateau, which lasts for approximately 10 s, was observed for the ARPC3 positive events (Fig. 5b). Based on these observations and previous experimental and computational modeling data[4,6,7], we propose that this plateau in branched actin-positive CME events represents stalled membrane bending due to an unfavorable local membrane environment.

We next tested the hypothesis that the asymmetric actin network might affect the lateral movements of endocytic coats on the plasma membrane. Interestingly, the ARPC3 positive CME sites showed significantly slower, but greater directional lateral movement before vesicle scission compared to the ARPC3 negative CME sites (Fig. 5c, d). After scission both ARPC3 positive and negative vesicles showed fast, apparently random movements (Fig. 5c, d). These data suggest that the asymmetric actin can stabilize the forming endocytic coat while pushing it in the plane of the plasma membrane with a lateral directional force.

To test the function of actin network assembly on CME, we treated the cells with an Arp2/3 inhibitor, CK666. We observed CME dynamics immediately after the treatment to minimize non-specific side effects that might be caused by prolonged actin assembly disruption. Even with moderate inhibition of actin network assembly at CME sites, indicated by reduced ARPC3 intensity (Supplementary Fig. 6a), we detected a small but significant increase in CME lifetimes (Supplementary Fig. 6b). This result is consistent with results from previous studies in SKMEL cells[7,12] and supports the hypothesis that Arp2/3-mediated actin assembly facilitates CME.

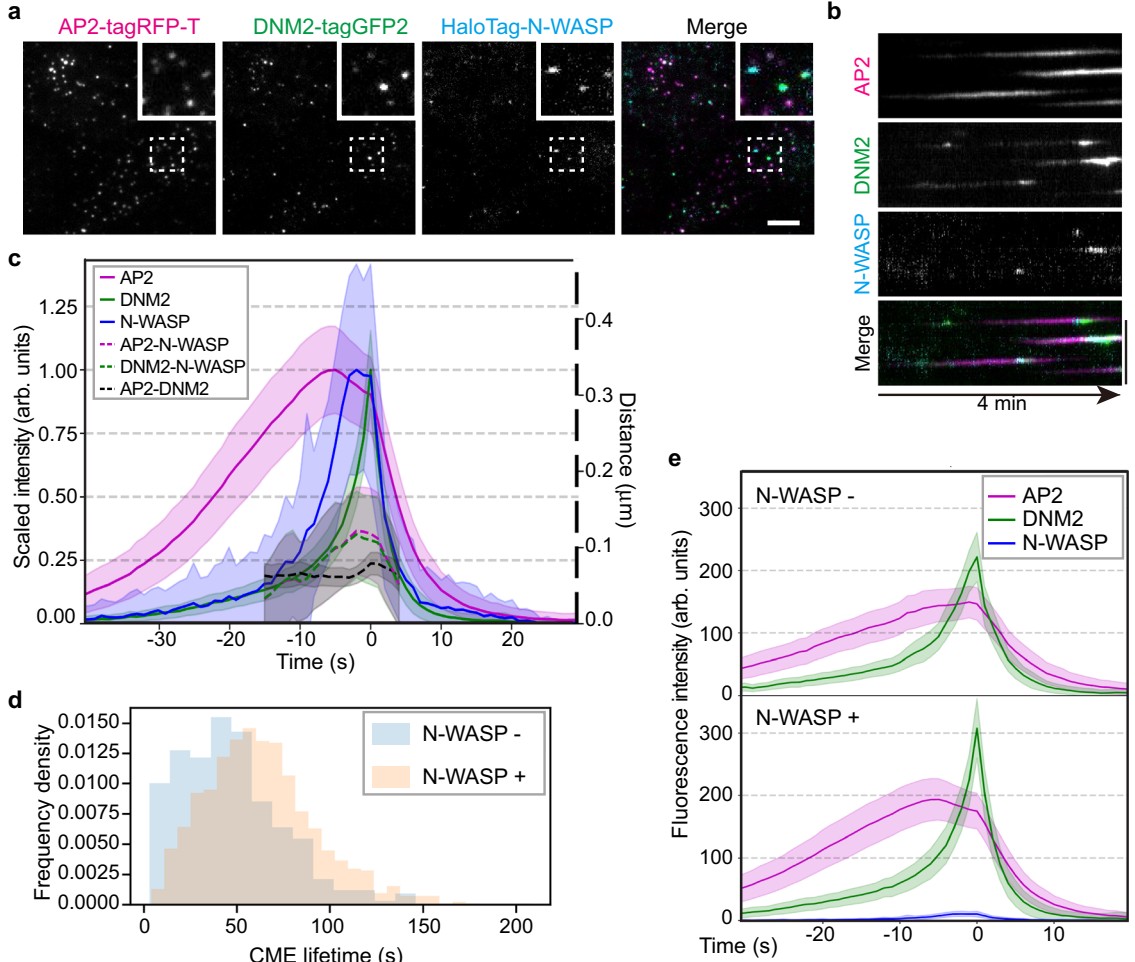

**Fig. 6 Asymmetric N-WASP recruitment to stalled CME sites. a** A representative single time frame image of a TIRF movie (Supplementary Movie 5) of AP2M1-tagRFP-T (magenta), DNM2-tagGFP2 (green) and JF635 ligand-conjugated HaloTag-N-WASP (cyan) in ADW cells. The highlighted region is boxed by a dashed line. Scale bar: 5 μm. **b** A representative kymograph of CME sites in ADW cells over a 4 min movie. Scale bar: 5 μm. **c** Averaged intensity (solid line) and distance (dashed line) vs time plots of N-WASP positive CME sites in ADW cells. Events are aligned to the frames showing the maximum DNM2 intensity. Intensity is scaled to 1 at peaks for each channel. Error bar: ¼ standard deviation. **d** N-WASP positive CME events have longer lifetimes. p-value from two-sided Welch's t-test: 9.06e-20. **e** Intensity vs time plots of averaged N-WASP negative (top) and positive (bottom) CME sites in ADW cells. Events were aligned to the frames showing the maximum DNM2 intensity. Error bar: ¼ standard deviation. Source data are provided in the Source Data file. **c**, **d** N-WASP negative CME sites: $N = 299$, N-WASP positive CME sites: $N = 1,199$.

**N-WASP is recruited asymmetrically to stalled CME sites**. To further explore how asymmetric assembly of actin networks at CME sites is regulated, we endogenously tagged N-WASP, an actin nucleation promoting factor (NPF) that plays roles in CME, in AP2M1-tagRFP-T/ DNM2-tagGFP2 genome-edited iPSCs (hereafter referred to as ADW cells, Fig. 6a, Supplementary Fig. 7a and Supplementary Movie 5). Quantitative imaging of budding yeasts demonstrated that initiation of productive actin assembly at CME sites requires the accumulation of yeast WASP and WIP (WASP Interacting Protein) to a certain level[37]. In our genome-edited iPSCs, we observed that N-WASP is recruited asymmetrically to CME sites mostly at the late stage of CME (Fig. 6b, c and Supplementary Fig. 7b, c). Longer lifetimes and a plateau in the AP2 intensity vs time plot were observed specifically for the N-WASP positive CME events (Fig. 6d, e), similar to the ARPC3 positive events (Fig. 5a, b). These data indicate that asymmetric NPF recruitment underlies the asymmetric architecture of branched actin networks at CME sites.

## Discussion

Using unbiased analysis of thousands of CME sites in unperturbed live cells, our study demonstrates that in mammalian cells clathrin coat assembly dynamics predict which sites will assemble actin, and show that at apparently stalled sites, actin assembles asymmetrically to facilitate successful vesicle formation.

Based on the data presented here, we propose an updated model for actin assembly at mammalian CME sites in which, beyond global tension-dependent changes in the requirement for actin assembly, highly localized differences give rise to heterogeneity even within the same patch of plasma membrane in the same cell (Fig. 7): (1) Where the local conditions are favorable for membrane deformation by coat proteins (Fig. 7 upper scenario), the membrane can undergo flat-U-Ω shape transitions in a relatively short time without actin assembly. When the coat grows large enough to form a Ω-shaped bud, sufficient dynamin can be recruited to perform scission, and there is little delay between coat expansion and scission; (2) Where the local conditions are not favorable, presumably under high membrane tension or other

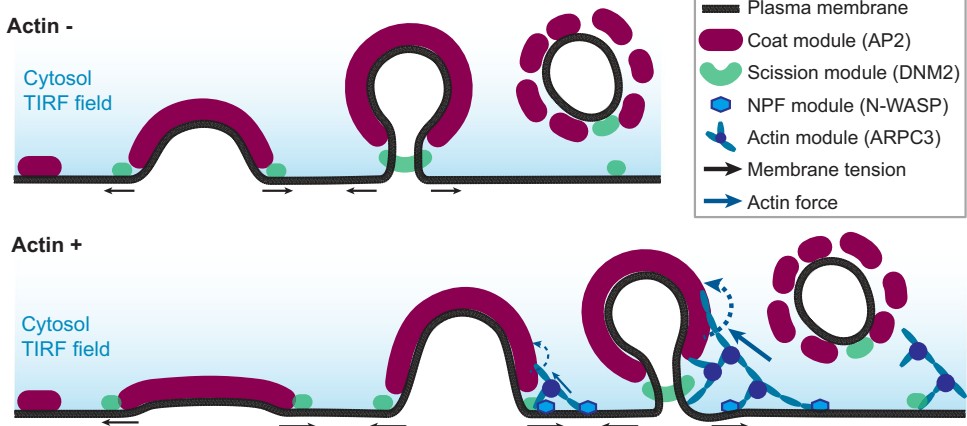

**Fig. 7 An updated schematic model of actin-negative and actin-positive clathrin-coated pits in human cells.** Actin assembly is induced at stalled CME sites, where asymmetric forces pull, bend and possibly twist the plasma membrane against membrane tension to drive membrane invagination and vesicle scission.

impediments, the coat protein-membrane interaction does not generate sufficient force to curve the membrane (Fig. 7 lower scenario). Here, extra force generation from actin assembly is required[4,6,10]. Asymmetric N-WASP recruitment activates actin nucleation mostly at one side of the clathrin coat, generating an asymmetric force that pulls the membrane into the cell with a similar action to a bottle cap opener. We speculate that this asymmetrical force contributes to asymmetric membrane deformation at endocytic sites observed by high-speed atomic force microscopy[38] and may act with dynamin[39] to twist the clathrin pit to promote scission at the neck. CME events with associated actin assembly have longer lifetimes, likely due to a delay between the end of coat expansion and progress toward vesicle scission, requiring adaptive recruitment of actin regulators followed by actin network assembly and membrane remodeling. This result establishes that site-to-site heterogeneity in actin dependence and involvement can be observed without manipulating actin assembly.

Our model provides further insights into the basis for inconsistent effects of actin drugs on CME[6,14,18,40–45]. Actin plays crucial roles in membrane shaping, cell adhesion, and membrane tension. Global disruption of actin dynamics is expected to dramatically change membrane tension and the available pool of actin and associated proteins and therefore to have both direct and indirect effects on CME. Here, we focused on in-depth analysis of the unperturbed process and detected preference for actin assembly at stalled CME events.

We used genome-edited ARPC3-HaloTag protein expressed at endogenous level as a marker of branched actin networks. This fluorescent protein allowed us to study the spatiotemporal dynamics of Arp2/3-nucleated actin structures at CME sites specifically (Fig. 1a, d). A large population of unbranched cortical actin filaments at CME sites has been observed by cryo-EM[46] and distinct effects of cortical actin-based structures on CME have been reported[36]. Previous studies on actin's spatiotemporal dynamics and function at CME sites applied actin probes such as Lifeact, which label actin filaments nucleated by different mechanisms, and which can perturb dynamic actin properties, possibly complicating conclusions reached in those studies[14,31,47,48].

WASP family proteins play a central role in regulation of Arp2/3-mediated actin assembly both spatially and temporally during various cellular processes. We observed that N-WASP is recruited asymmetrically to stalled CME sites (Fig. 6). We propose this asymmetric recruitment underlies the asymmetric actin

architecture at CME sites. Quantitative live-cell imaging studies in yeast suggested concentration of yeast WASP at CME sites through multivalent protein interactions is important for robust, switch-like actin assembly at CME sites[37]. Future modeling together with quantitative in vivo and in vitro studies are needed to determine whether concentrating N-WASP into a small space, which is asymmetrically located relative to CME sites, is a more efficient way to generate sufficient actin force for productive CME, compared to recruiting more N-WASP into a larger space that encircles the CME sites. Future computational modeling studies on how asymmetric actin network assembly provides forces and generates a torque for vesicle formation and membrane remodeling will deepen our understanding of actin's functions in a host of actin-mediated processes.

While this study provides evidence that actin assembles asymmetrically at mammalian CME sites, some questions remain open. For example, what is the molecular mechanism that couples recruitment of N-WASP and other actin assembly factors to the stalled CME sites? Many factors, including membrane tension[6,9,10], cell cycle[9], cell adhesion[8,49] and cargo size and composition[32,33] appear to contribute to the actin requirement for CME. How do these different factors regulate the actin assembly machinery? Answering these questions in the future will deepen our understanding of dynamic actin assembly regulation in membrane trafficking.

## Methods

**Cell culture**. The WTC10 hiPSC line was obtained from the Bruce Conklin Lab at UCSF. hiPSCs were cultured on Matrigel (hESC-Qualified Matrix, Corning) in StemFlex medium (Thermo Fisher) with Penicillin/ Streptomycin in 37 °C, 5% CO$_2$. Cultures were passaged with Gentle Cell Dissociation reagent (StemCell Technologies, Cat#: 100–0485) twice every week.

**Genome-editing**. The AP2M1 gene was edited in WTC10 hiPSCs as previously described using TALENs targeting exon 7 of the AP2M1 gene[50]. Both alleles of AP2M1 were tagged with tagRFP-T. The Cas9-crRNAtracrRNA complex electroporation method was used sequentially to edit DNM2 and ARPC3 gene in AP2M1-tagRFP-T genome-edited hiPSCs, as previously described[12,18]. The same method was used to edit the WASL gene in AP2M1-tagRFP-T/DNM2-tagGFP2 genome edited hiPSCs. *S. pyogenes* NLS-Cas9 was purified in the University of California Berkeley QB3 MacroLab. TracrRNA and crRNA that target CCTGCTCGAC-TAGGCCTCGA (DNM2), CCTGGACAGTGAAGGGAGCC (ARPC3) and AGCTCATGGTTTCGCCGGCG (WASL), were purchased from IDT. Gibson assembly (New England Biolabs) was used to construct donor plasmids containing DNM2 5′ homology-ggtaccagtggcggaagc-tagGFP2-DNM2 3′ homology, ARPC3 5′ homology-ggatccggtaccagcgatccaccggtcgccacc-HaloTag-ARPC3 3′ homology, and WASL 5′ homology-HaloTag-agcgatccaccggtcgccaccggatcc-WASL 3′ homology sequences, respectively. Three days after electroporation (Lonza, Cat#: VPH-5012)

of the Cas9-crRNA-tracrRNA complex and donor plasmid, the tagGFP2 or HaloTag positive cells were single cell sorted using a BD Bioscience Influx sorter (BD Bioscience) into Matrigel-coated 96-well plates. Clones were confirmed by PCR and Sanger sequencing of the genomic DNA locus around the insertion site. Both alleles of DNM2 and ARPC3 were tagged with tagGFP2 and HaloTag, respectively, and one allele of WASL was tagged with HaloTag in the hiPSC lines used in this study.

**Western blotting**. Cells were dissociated from the well using Gentle Cell Dissociation reagent (StemCell Technologies, Cat#: 100-0485). Total proteins were extracted by adding 1 ml of cold 10% TCA to the cell pellets, incubated on ice for 30 min, and spun down by centrifuging at 4 °C, 13400 × $g$ for 10 min. Protein pellets were dissolved in loading buffer (50 mM HEPES, pH 7.4, 150 mM NaCl, 1 mM MgCl2, 5% BME, 5 mM DTT and protease inhibitor) and loaded onto an acrylamide gel for SDS-PAGE and transferred to nitrocellulose membranes for immunoblotting. Blots were incubated overnight at 4 °C with primary antibodies targeting Tag(CGY)FP (1:2000 dilution in 2% milk, Evrogen, Cat#: AB121), HaloTag (1:2000 dilution in 2% milk or 1:1000 dilution in 0.5% milk, Promega, Cat#: G9211), GAPDH (1:100,000 dilution in 0.5% milk, Proteintech, Cat#: 10494-1-AP), respectively, and subsequently incubated in the dark at room temperature for 1 hr with secondary antibodies.

**TIRF live-cell imaging**. Two days before imaging, hiPSCs were seeded onto Matrigel-coated 4-well chambered cover glasses (Cellvis). Halotag was labeled by JF635- HaloTag ligand[51]. Cells were incubated in StemFlex medium with 100 mM JF635-HaloTag for 45 min and the unbound ligands were washed away by three washes with 5 min incubation in prewarmed StemFlex medium. Cells were imaged on a Nikon Ti-2 inverted microscope fitted with TIRF optics and a sCMOS camera (Hamamatsu). Cells were maintained at 37 °C with a stage top incubator (OKO Lab) in StemFlex medium with 10 mM HEPES. Images were acquired with Nikon Elements. Channels were acquired sequentially at a 1 s interval and 300 ms exposure time over 4 min.

For the CK666 treatment experiments, a 50 mM CK666 (SML0006, Sigma Aldrich) stock solution was prepared in DMSO and kept at −80 °C. 200 μM CK666 solution and 4% DMSO (v/v) solution for a control were prepared fresh in StemFlex medium with 10 mM HEPES and incubated at 37 °C prior to use. Live-cell imaging was performed 1 min after adding an equal volume of medium containing CK666 or DMSO into the imaging chamber to achieve a final concentration of 100 μM CK666 or 2% DMSO.

**TIRF image processing and analysis**. Events (i.e., tracked diffraction-limited spots) were extracted from cmeAnalysis[30] and processed in Python Jupyter Notebooks. AP2-tagRFP-T was used as the fiducial marker for clathrin-mediated endocytosis in cmeAnalysis tracking experiments. DNM2 was used as a secondary channel to mark vesicle scission and the termination of vesicle formation. ARPC3-HaloTag was tracked separately from paired AP2/DNM2 movies and linked to CCPs downstream of cmeAnalysis to allow for determining discrete ARPC3 nucleation and disassembly events.

Four generalized processing steps were applied to identify clathrin-coated pits (AP2 tracks) with single DNM2 peaks: track feature abstraction, feature dimensionality reduction, event clustering, and DNM2-peak detection. First, tracks that are defined by fitted positions and intensities for single events were generated using cmeAnalysis. Then, AP2 and DNM2 tracks were decomposed into dynamic features describing the dynamics of the position and brightness for each event. Each tracked event, which was once an arbitrary array of intensities and positions, was now a discrete vector of fixed length. The mapping of each track to discrete features was done to generalize the dynamics of heterogeneous tracked events into a set of interpretable coordinates.

Following feature abstraction, the output array is a 2-dimensional matrix with N rows (N tracked events) and M columns (M discrete features per track). These features were individually scaled to normal distributions to remove the variability in scale and dampen the effects of outliers. For instance, the 'lifetime' feature (AP2 lifetime) ranged from a few seconds to several minutes on a scale of seconds, whereas the 'DNM2-peak fraction' feature (where the DNM2 peak is located within one AP2 event) ranges from 0 to 1. Following feature re-scaling, these events, which each contain over thirty features, were projected to a lower-dimensional space via principal component analysis. The derived clusters were separated using a Gaussian Mixture Model and events were assigned to clusters based on their highest probability of identity to one cluster.

DNM2-positive events represented a distinct cluster of tracks that had detectable DNM2 throughout the event, were long lived, and were below the threshold of motility expected for transient, non-CME-derived clathrin-coated vesicle "visitors"[50] to the TIRF field. To characterize a single-DNM2 peak, DNM2-positive events were surveyed over a range of values set for the minimum DNM2 peak height, width, and peak-to-peak temporal distance. For each peak-defining parameter combination, all DNM2-positive events were categorized as having zero, one, or two or more peaks. After finding single-peaked events in a fixed peak-parameter combination, the lifetime distribution of lifetimes for single peak events was fit to the expected underlying distribution, a Rayleigh distribution[52], where the

best-fitting parameter combination was chosen to identify single-peaked events. Single DNM2-peaked events were kept as CME sites for the remainder of the analysis. All code associated with this analysis, generating Figs. 4–6, and a detailed step-by-step protocol, are available at https://github.com/DrubinBarnes/Jin_Shirazinejad_et_al_branched_actin_manuscript.

**Two-color 3D STORM imaging and analysis**. 12 mm round coverslips were sonicated in distilled water and sterilized for 20 min in 70% ethanol, air-dried and coated with Matrigel in 24-well plates. Cells were seeded onto Matrigel-coated coverslips two days before fixation. For clathrin and actin two-color imaging, cells were fixed first for 1 min in 0.3% (v/v) glutaraldehyde (GA) solution containing 0.25% (v/v) Triton in cytoskeleton buffer (CB: 10 mM MES, 150 mM NaCl, 5 mM EGTA, 5 mM Glucose, 5 mM MgCl2, 0.005% NaN3, pH 6.1) and then immediately fixed for 20 min in 2% (v/v) GA solution in CB. Both solutions were prepared fresh from a 10% GA stock (Electron Microscopy Science, Cat#: 16120). After fixation, samples were incubated twice for 5 min in freshly prepared 0.1% (w/v) NaBH4 in PBS. For clathrin and ARPC3-HaloTag imaging, cells were fixed for 20 min in 4% (v/v) PFA (Electron Microscopy Sciences, Cat#: 15710) in CB. Subsequently, both types of samples were washed 3 times for 10 min in PBS. Samples were then blocked for 20 min in blocking buffer [3% (w/v) BSA and 0.1% (w/v) Saponin in PBS]. Clathrin light chain (Invitrogen, Cat#: MA5-11860, 1:200 dilution) and Halotag (Promega, Cat#: G9281, 1:200 dilution) antibodies were used in blocking solution. Primary antibody immunostaining was performed overnight at 4 °C. On the next day, samples were washed three times in washing buffer (0.1× blocking buffer in PBS) for 10 min. Samples were incubated with secondary antibody in blocking buffer for 30 min at room temperature in the dark and were washed three times for 10 min in washing buffer, and then three times for 10 min in PBS. Mouse secondary antibody (Jackson ImmunoResearch, Code#: 715-005-151) conjugated with CF680 (Biothium, Cat#: 92139) (1:50) was used to stain clathrin and actin samples. Commercial mouse secondary antibody-AF647 (ThermoFisher, Cat#: A32787; 1:400) and rabbit secondary antibody (Jackson ImmunoResearch, Code#: 711-005-152) conjugated with CF680 (Biothium, Cat#: 92139) (1:50) were used to stain the clathrin and ARPC3-HaloTag. Clathrin and actin samples were then stained with 0.5 μM Phalloidin-AF647 (Fisher Scientific, Cat#: A22287) in PBS and kept at room temperature in the dark for 2 h. Samples were washed three times with PBS before STORM imaging.

STORM imaging was performed as previously described on a homebuilt STORM setup[7,53]. Samples labeled by AF647 and CF680 were excited by an 647 nm laser. The emission of both AF647 and CF680 was then split into two light paths as two channels using a dichroic mirror (Chroma, Cat#: T685lpxr), and each channel was projected onto one-half of an EMCCD camera (Andor iXon Ultra 897). Color assignment of each localization was based on its intensity in the two channels. A cylindrical lens was inserted into the transmitted channel to acquire 3D localization[28]. 3D position of each localization was determined from the ellipticity of each point spread function.

The raw STORM data was processed according to previously described methods[28,54] and single-molecule localization and optical reconstruction were performed using the Insight3 software (developed by Dr. Bo Huang at University of California, San Francisco and Dr. Xiaowei Zhuang at Harvard University). To quantify the distances between centroids of clathrin and actin or ARPC3 signals, we first manually cropped regions that contain single CME sites associated with actin or ARPC3 structures using Insight3 software. Then we identified and saved the xy positions of single-molecule localizations in cropped regions as txt files. We next calculated the centroid of signals by averaging xy positions of each channel and calculated the distance between the centroids of two channels using MATLAB.

**Airyscan imaging and processing**. Live cell sample preparation was performed as described in "TIRF live-cell imaging". Cells were imaged on a Zeiss LSM 900 inverted microscope using an Airyscan 2 detector and Multiplex 4Y line scanning mode. Cells were maintained at 37 °C, 5% CO2 in StemFlex medium. Images were acquired and processed using the ZEN 3.1 system. Channels were acquired sequentially for each line with 0.2 s/frame intervals over 3 min. Images were processed with 3.7 deconvolution strength 2D Airyscan processing. Alignment between channels was corrected using fluorescent bead images.

**Reporting summary**. Further information on research design is available in the Nature Research Reporting Summary linked to this article.

## Data availability
The data that support this study are available from the corresponding author upon reasonable request. Source data are provided with this paper.

## Code availability
The Jupyter Notebooks used for live-cell imaging analysis can be found at https://doi.org/10.5281/zenodo.6575159.

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

## Acknowledgements

M.J. was funded by American Heart Association Postdoctoral Fellowship (18POST34000029). D.G.D. was funded by NIH MIRA grant R35GM118149. K.X. is a

Chan Zuckerberg Biohub investigator and acknowledges support from NIH (DP2GM132681). S.U. is a Chan Zuckerberg Biohub Investigator and acknowledges support from the Philomathia Foundation, and Chan Zuckerberg Initiative Imaging Scientist program. We would like to thank Dr. Yidi Sun and Dr. Matthew Akamatsu for insightful comments on the manuscript; the Conklin Lab at UCSF for providing the WTC10 human iPSC line; the Lavis Lab at Janelia Research Campus for providing JF635 HaloTag ligand; Dr. Sun Hae Hong for generating the AP2-tagRFP-T iPSC line; the UC Berkeley QB3 MacroLab for purified *S. pyogenes* NLS-Cas9; the Luo Lab at UC Berkeley for sharing their electroporator and the UC Berkeley Cancer Research Laboratory Flow Cytometry Facility for iPSC sorting.

## Author contributions

M.J., C.S. and D.G.D. conceived the study and experiments. M.J. and A.Y. generated the genome-edited cell lines. M.J. performed live cell data acquisition and sample preparation for super-resolution microscopy. B.W. and K.X. performed super-resolution microscopy and super-resolution data reconstruction. C.S. developed computational analysis tools and S.U., J.S., D.G.D. and M.J. supported the data analysis. M.J., C.S., and D.G.D. prepared the figures and wrote the manuscript with feedback from the other authors.

## Competing interests

The authors declare no competing interests.
