## [Peer Review File · Nature Communications]

Branched actin networks are organized for asymmetric force production during clathrin-mediated endocytosis in mammalian cellsReviewers' Comments:

Reviewer #1:

Remarks to the Author:

The authors revisit here the idea that in mammalian cells some CME sites, but not all, display actin polymerization concomitant with the scission step. While controversial in the past, a consensus in the field is emerging that a subset of CME sites in mammalian cells thus in fact utilize actin polymerization likely to overcome some barrier such as membrane tension. This is certainly a worthwhile topic to study. In this regard the paper confirms and extends the observations previously published by this group with gene edited cells (Grassart et al, JCB 2014) and others with overexpressed proteins (Boulant et al, NCB 2011; Li et al, Science 2015, etc). The key new data here is that CME sites decorated with Arp2/3 correlate with a longer lifetime of coat protein AP2. This is interpreted as actin polymerization "rescuing" stalled sites, which is a plausible and logical idea. However, this reviewer believes most of the novelty of this manuscript comes from the data suggesting the branched actin polymerization machinery (Arp2/3, N-WASP) does not distribute symmetrically around the CME site but instead is localized to one side only. This is in contrast to the situation in yeast, which has a ring of Las17 (yeast WASP) at the base. These results will have significant implications for the membrane trafficking and actin fields and clearly worthy of publication. Nevertheless, there are some aspects of this manuscript that could be enhanced before publication.

1) As mentioned above, the asymmetric distribution of the actin polymerization machinery is the more novel aspect of this work. However, the superresolution data was obtained with fixed cells and the live cell data with diffraction limited microscopy. If feasible, it would be ideal to analyze live cells using superresolution microscopy to further support this point.

2) Figure 1 shows nice examples of asymmetrically distributed actin (phalloidin) and Arp2/3 complex (ARPC3) relative to clathrin. However, it would be important to show some form of quantification of these data. Is this a common or rare occurrence? What percentage of CME sites show off-centered actin?

3) In the analysis of Figure 2, 3, etc, it would be important to include what percentage of the total detected structures were analyzed as discrete CME sites (and what percentage were plaques and splitting events, which are not analyzed).

4) The idea that actin polymerization rescues stalled sites is attractive. Correlation shown in Figure 4a, b is consistent with this idea. However, Figure 3a, late actin (1,385 sites) shows that even in the short lived AP2 sites (no plateau) Arp2/3 shows up robustly. This goes against the idea that actin polymerization rescues stalled sites. Could the authors comment on this apparent discrepancy with the model?

5) In contrast to the results shown here, Li et al, Science 2015, found CME sites associated with filamentous actin were slightly shorter lived than those not associated with actin. Could the authors offer an explanation for the different results?

6) There is no real experimental connection here between the CME sites showing actin polymerization and membrane tension. This connection is speculative and could be shored up, for instance if an increase in tension (such as using hypotonic media) brings about a higher proportion of Arp2/3-labelled CME sites. Otherwise, the whole conclusion/discussion about tension could be toned down. Related to this point, could other factors besides membrane tension (such as cargo load/type) be at play in determining the requirement for actin?

7) The discussion about constant coat vs constant curvature models appears exaggerated given that these models are not really tested by the experiments presented in this work.

Minor:

- References 6 and 11 are the same.
- Line 45: " that nature of which are obscure" should read "the nature of which is obscure"
- Line 147: "Extended Data Fig. 3d." should be "Supplementary Fig. 3d."
- Line 285-286: "Perhaps the constant area model applies primarily for actin-negative sites and the constant area model applies..." sentence needs revision.

Reviewer #2:

Remarks to the Author:

In this article, Jin et al. generated triple-edited iPS cells to follow the role of actin during CME using TIRF. Using an advanced image analysis pipeline, they report that ARPC3 is asymmetrically recruited to stalling clathrin-coated pits. Many of their observations appear to be of a confirmatory nature (not a bad thing, but I will let the editor decide if novelty is an issue here). In its current form, the manuscript feels a bit too preliminary to recommend publication.

My two main concerns are outlined below:

The (temporal and spatial) resolution of the TIRF microscope used for most of this study is not adapted to what the authors want to observe. The authors try to validate the absence of imaging artifacts, but a large part of the issue remains. I would encourage the authors to repeat some of their live imaging experience using TIRF-SIM / STED or any other suitable super-resolution modality.

One of the main findings from this manuscript is that asymmetric Arp2/3-mediated actin networks rescue stalled CME (this is the title). However, the role of asymmetric Arp2/3 recruitment to CME was not tested at all, only observed. Without further experimentation, it is not possible to conclude this. What is the role of Arp2/3 in stalled CME? what is the consequence of disrupting the asymmetric recruitment? I understand this is a lot of work, but then if not functional data can be obtained, the authors should change the title, acknowledge that correlation is not the same as causation, and tone down their conclusion throughout the manuscript.

Minor comments

I would recommend the author rewrite the abstract to make it more informative. Sentences such as "were analyzed using three-dimensional (3D) super-resolution microscopy, live-cell imaging, and machine-learning-based computation" while full of buzz words are not very helpful for the reader.

The authors should label the SI movies to make them easier to understand.

Observation made in Figure 1 should be quantified.

The differences between the curves obtained in Fig3-5 should be quantified and statistically compared.

Figure 3 is very hard to read. Perhaps the authors could find a way to better highlight the differences between the actual data and the randomized data. And between the early actin and late actin.

The material and methods are incomplete:

The STORM data processing and analysis is not described

The "TIRF image processing" part of the material and methods is very hard to understand. I would suggest the authors spend a bit of time explaining their image analysis pipeline in greater detail.

Response to referees

Reviewer #1 (Remarks to the Author):

The authors revisit here the idea that in mammalian cells some CME sites, but not all, display actin polymerization concomitant with the scission step. While controversial in the past, a consensus in the field is emerging that a subset of CME sites in mammalian cells thus in fact utilize actin polymerization likely to overcome some barrier such as membrane tension. This is certainly a worthwhile topic to study. In this regard the paper confirms and extends the observations previously published by this group with gene edited cells (Grassart et al, JCB 2014) and others with overexpressed proteins (Boulant et al, NCB 2011; Li et al, Science 2015, etc). The key new data here is that CME sites decorated with Arp2/3 correlate with a longer lifetime of coat protein AP2. This is interpreted as actin polymerization “rescuing” stalled sites, which is a plausible and logical idea. However, this reviewer believes **most of the novelty of this manuscript comes from the data suggesting the branched actin polymerization machinery (Arp2/3, N-WASP) does not distribute symmetrically around the CME site but instead is localized to one side only.** This is in contrast to the situation in yeast, which has a ring of Las17 (yeast WASP) at the base. These results will have significant implications for the membrane trafficking and actin fields and clearly worthy of publication. Nevertheless, there are some aspects of this manuscript that could be enhanced before publication.

We very much appreciate the enthusiasm of this reviewer for our study and for indicating that “These results will have significant implications for the membrane trafficking and actin fields and clearly worthy of publication”.

1) As mentioned above, the asymmetric distribution of the actin polymerization machinery is the more novel aspect of this work.

We modified our title to emphasize this novel aspect of the work.

However, the superresolution data was obtained with fixed cells and the live cell data with **diffraction limited microscopy**. If feasible, it would be ideal to analyze live cells using superresolution microscopy to further support this point.

This point was made by both reviewers of the manuscript and we have now endeavored to address the issue. We do wish to make the point that our original analysis did include TIRF imaging of live cells that supported the STORM data when we tracked the centroids of the fluorescence signals as a function of time. Nevertheless, we have now added additional analysis in which we employed Zeiss Airyscan2 microscopy to analyze the distribution of the Arp2/3 complex (ARPC3) relative to dynamin (DNM2) in live cells (new Figure 3). In these studies, we were able to achieve sub-diffraction-limit spatial resolution and 200 ms/frame temporal resolution. Our observations from Airyscan2 imaging of live cells are entirely congruent with, and supportive of, ones we made previously with STORM imaging of fixed cells and TIRF imaging of live cells. Thus, three different imaging modalities have now all resulted in the same conclusions.

2) Figure 1 shows nice examples of asymmetrically distributed actin (phalloidin) and Arp2/3 complex (ARPC3) relative to clathrin. However, it would be important to show some form of quantification of these data. Is this a common or rare occurrence? What percentage of CME sites show off-centered actin?

We appreciate this suggestion and have now added quantification of the STORM data. We present a histogram in which we have quantified the distribution of distances between the centroid of the clathrin and actin or Arp2/3 complex (ARPC3) signals in 67 and 161 sites, respectively (Figure 1. c, f).

This new quantification data has strengthened our manuscript by showing that asymmetric actin assembly at CME sites is the norm, not the exception.

3) In the analysis of Figure 2, 3, etc, it would be important to include what percentage of the total detected structures were analyzed as discrete CME sites (and what percentage were plaques and splitting events, which are not analyzed).

We adapted *cmeAnalysis* (Aguet et al, *Dev. Cell*, 2013, <https://doi.org/10.1016/j.devcel.2013.06.019>) tracking for our analysis. We added the requested percentages in the Supplemental Fig 3. To our knowledge, our filtering of CME site splitting and merging events is consistent with established practices for how other investigators have dealt with these events in their publications. Our additional, automated filtering steps are now well documented in Supplemental Fig. 3. Here how many sites were eliminated, and how many were analyzed, are now transparently presented. About 4.3% of total events captured were quantified.

4) The idea that actin polymerization rescues stalled sites is attractive. Correlation shown in Figure 4a, b is consistent with this idea. However, Figure 3a, late actin (1,385 sites) shows that even in the short lived AP2 sites (no plateau) Arp2/3 shows up robustly. This goes against the idea that actin polymerization rescues stalled sites. Could the authors comment on this apparent discrepancy with the model?

We appreciate this opportunity to clarify an apparent discrepancy. We see that there is a short plateau in the 40s-60s cohort. However, there is no obvious plateau in the <40s cohort, which only contains less than 8% of the events of the whole population. Moreover, as described here, we believe that many or most of the actin associations for these short-lived sites are likely to be “false positives”. As shown in Supplementary Fig. 4b (new), late-stage actin “assembly” was detected at more than 12% of CME sites even in the randomized dataset, which was generated by pairing ARPC3 images with AP2 and DNM2 images from an unrelated movie (Supplementary Fig. 4a), suggesting many or all actin assembly detections at short-lived AP2 sites may be caused by random association of nearby actin with the CME sites. In such cases, spurious associations are less likely to be classified as such because the apparent actin association is present for a longer fraction of the AP2 lifetime than in the longer lifetime cohort. Because it is more difficult to distinguish between CME-specific actin assembly (late actin) and coincident actin appearance around CME sites (early actin) for the events with shorter CME lifetimes, these data are more likely to include false-positive Arp2/3+ events than the cohorts with longer AP2 lifetimes.

5) In contrast to the results shown here, Li et al, *Science* 2015, found CME sites associated with filamentous actin were slightly shorter lived than those not associated with actin. Could the authors offer an explanation for the different results?

There are several differences between these studies that might explain this discrepancy. For one thing, Li et al use LifeAct to visualize actin, while we used an endogenously-tagged Arp2/3 subunit. Lifeact can affect actin spatiodynamics, membrane tension, cell morphology, behavior and function (Xu and Du, *Front. Cell Dev. Biol.*, 2021, <https://doi.org/10.3389/fcell.2021.746818>) (Flores et al., *Sci Rep*, 2019, <https://doi.org/10.1038/s41598-019-40092-w>). Our analysis employed genome-edited ARPC3, which labels branched actin networks specifically and not actin generated by formins, for example. These differences and the use of different cell types could all potentially contribute to the different observations. (See the last paragraph on Page13)

6) There is no real experimental connection here between the CME sites showing actin polymerization and membrane tension. This connection is speculative and could be shored up, for instance if an increase in tension (such as using hypotonic media) brings about a higher proportion of

Arp2/3-labelled CME sites. Otherwise, the whole conclusion/discussion about tension could be toned down. Related to this point, could other factors besides membrane tension (such as cargo load/type) be at play in determining the requirement for actin?

We performed the suggested hypotonic medium experiment. We observed that the number and intensity of ARPC3 structures increased after the hypotonic treatment (see figure below). However, due to crowding of the ARPC3 signals in these cells, it was not possible to accurately track and associate ARPC3 structures to the CME sites. Therefore, we toned down our title, conclusions and discussion about tension and added more discussion about possible factors, including membrane tension and cargo, that may stall the CME process.

7) The discussion about constant coat vs constant curvature models appears exaggerated given that these models are not really tested by the experiments presented in this work.

We removed that part of the Discussion.

Minor:

- References 6 and 11 are the same. **Fixed**
- Line 45: “ that nature of which are obscure” should read “the nature of which is obscure” **Fixed**
- Line 147: “Extended Data Fig. 3d.” should be “Supplementary Fig. 3d.” **Fixed**
- Line 285-286: “Perhaps the constant area model applies primarily for actin-negative sites and the constant area model applies...” sentence needs revision. **We removed that part of the Discussion.**

Reviewer #2 (Remarks to the Author):

In this article, Jin et al. generated triple-edited iPS cells to follow the role of actin during CME using TIRF. Using an advanced image analysis pipeline, they report that ARPC3 is asymmetrically recruited to stalling clathrin-coated pits. Many of their observations appear to be of a confirmatory nature (not a bad thing, but I will let the editor decide if novelty is an issue here). In its current form, the manuscript feels a bit too preliminary to recommend publication.

We do not believe that this work is confirmatory but is the first to address this matter in an unbiased, systematic manner. Knowing how actin is organized at CME sites is foundational for understanding the force-generation mechanism. The conclusion that actin assembles at CME sites asymmetrically is not a foregone conclusion as this feature is drastically different from what has been observed in the very well-studied budding yeast CME process. As Reviewer 1 commented, our quantitative demonstration that actin assembles at CME sites asymmetrically is a novel aspect of the work. More

importantly, modeling studies of actin assembly during mammalian CME have extrapolated from experimental observation using fixed cell imaging approaches, and assume that the actin network initiates asymmetrically but then expands to encircle the CME site. Our unbiased fixed-cell and live-cell data show that branched actin networks and N-WASP remain asymmetric at CME sites until successful scission.

My two main concerns are outlined below:

The (temporal and spatial) resolution of the TIRF microscope used for most of this study is not adapted to what the authors want to observe. The authors try to validate the absence of imaging artifacts, but a large part of the issue remains. I would encourage the authors to repeat some of their live imaging experience using TIRF-SIM / STED or **any other suitable super-resolution modality**.

Reviewer#1 had a very similar comment. Please see our response to point 1) of Reviewer #1. Furthermore, we provided new evidence that the displacement between AP2 and ARPC3 signals detected by TIRF imaging is not due to the chromatic aberration. Our new analysis shows that the separation we observed greatly exceeds any such chromatic aberration effects (Supplementary Fig. 5b).

One of the main findings from this manuscript is that asymmetric Arp2/3-mediated actin networks rescue stalled CME (this is the title). However, the role of asymmetric Arp2/3 recruitment to CME was not tested at all, only observed. Without further experimentation, it is not possible to conclude this. What is the role of Arp2/3 in stalled CME? what is the consequence of disrupting the asymmetric recruitment? I understand this is a lot of work, but then if not functional data can be obtained, the authors should change the title, acknowledge that correlation is not the same as causation, and tone down their conclusion throughout the manuscript.

We have now tested the role of the Arp2/3 complex in CME using the Arp2/3 inhibitor CK666. Our data show that Arp2/3 inhibition slows CME (Supplementary Fig. 6). This analysis does not address whether the asymmetry per se is what is important. To address this important question, we plan in future studies to address this point through mathematical modeling.

We changed our title to better emphasize our findings of asymmetric actin assembly at mammalian CME sites and removed mention of stalled CME sites.

Minor comments

I would recommend the author rewrite the abstract to make it more informative. Sentences such as “were analyzed using three-dimensional (3D) super-resolution microscopy, live-cell imaging, and machine-learning-based computation” while full of buzz words are not very helpful for the reader.

We were not attempting to impress the reader with buzz words, but to inform the reader about our methodology. Nevertheless, we have endeavored to make the Abstract more informative.

The authors should label the SI movies to make them easier to understand. **Done**

Observation made in Figure 1 should be quantified. **Done (same as the second comment of Reviewer #1).**

The differences between the curves obtained in Fig3-5 should be quantified and statistically compared. **Done.**

Figure 3 is very hard to read. Perhaps the authors could find a way to better highlight the differences between the actual data and the randomized data. And between the early actin and late actin. We now present a dot plot, which we hope satisfactorily addresses this concern (Supplementary Fig. 4).

The material and methods are incomplete:

The STORM data processing and analysis is not described. Added.

The “TIRF image processing” part of the material and methods is very hard to understand. I would suggest the authors spend a bit of time explaining their image analysis pipeline in greater detail.

Updated.

Reviewers' Comments:

Reviewer #1:

Remarks to the Author:

The authors have satisfactorily addressed the points raised in the original review. The following are a few minor issues that could be improved but do not require an additional round of review:

1. The Airyscan2 imaging data is useful to support the actin offset relative to the coat or dynamin. However, the experiment could be better described. For example, is the data shown in Figure 3 taken on the ventral side of the cell. (LSM 900 is a laser scanning confocal microscope)? The small area shown in Figure 3 (also Supp Video 3) evidently allowed a faster rate of acquisition but, if available, a picture of a larger area showing the entire cell would be helpful.
2. Supplementary videos are not described although they are referenced in the main text and legend to figures.
3. Update reference 46 (bioRxiv preprint) with Dev Cell article in press by the Drubin lab.

Reviewer #2:

Remarks to the Author:

The authors have addressed all my suggestions! Congratulation on a beautiful paper.

I recommend publication.

REVIEWERS' COMMENTS

Reviewer #1 (Remarks to the Author):

The authors have satisfactorily addressed the points raised in the original review. The following are a few minor issues that could be improved but do not require an additional round of review:

1. The Airyscan2 imaging data is useful to support the actin offset relative to the coat or dynamin. However, the experiment could be better described. For example, is the data shown in Figure 3 taken on the ventral side of the cell. (LSM 900 is a laser scanning confocal microscope)? The small area shown in Figure 3 (also Supp Video 3) evidently allowed a faster rate of acquisition but, if available, a picture of a larger area showing the entire cell would be helpful.
2. Supplementary videos are not described although they are referenced in the main text and legend to figures.
3. Update reference 46 (bioRxiv preprint) with Dev Cell article in press by the Drubin lab.

Reviewer #2 (Remarks to the Author):

The authors have addressed all my suggestions! Congratulation on a beautiful paper.

I recommend publication.

We would like to thank the two reviewers for their positive comments on our revised manuscript. We appreciate all of their suggestions on how to improve our manuscript. The remaining comments of Reviewer #1 are addressed below. Responding to these comments further improved our manuscript.

1. The Airyscan2 imaging data is useful to support the actin offset relative to the coat or dynamin. However, the experiment could be better described. For example, is the data shown in Figure 3 taken on the ventral side of the cell. (LSM 900 is a laser scanning confocal microscope)? The small area shown in Figure 3 (also Supp Video 3) evidently allowed a faster rate of acquisition but, if available, a picture of a larger area showing the entire cell would be helpful.

Yes, the data shown in Figure 3 were taken on the ventral membrane of the cells. We added this information to the figure legend. Also, we now show a larger area of the cells in Supplementary Movie 3.

2. Supplementary videos are not described although they are referenced in the main text and legend to figures.

The legends for the videos had been uploaded separately. We are sorry for the inconvenience caused by not adding a copy of the legends with the main files. Legends for the Supplementary Movies are now included.

3. Update reference 46 (bioRxiv preprint) with Dev Cell article in press by the Drubin lab.

Done.